# Learning from Offline Foundation Features with Tensor Augmentations

**Emir Konuk**[1,2], **Christos Matsoukas** [1,2], **Moein Sorkhei** [1,2] , **Phitchapha Lertsiravaramet** [1,2]

**Kevin Smith** [1,2]

[1] KTH Royal Institute of Technology, Stockholm, Sweden
[2] Science for Life Laboratory, Stockholm, Sweden
{ekonuk, ksmith}@kth.se

## Abstract

We introduce Learning from Offline Foundation Features with Tensor Augmentations (LOFF-TA), an efficient training scheme designed to harness the capabilities of foundation models in limited resource settings where their direct development is not feasible. LOFF-TA involves training a compact classifier on cached feature embeddings from a frozen foundation model, resulting in up to 37× faster training and up to 26× reduced GPU memory usage. Because the embeddings of augmented images would be too numerous to store, yet the augmentation process is essential for training, we propose to apply tensor augmentations to the cached embeddings of the original non-augmented images. LOFF-TA makes it possible to leverage the power of foundation models, regardless of their size, in settings with limited computational capacity. Moreover, LOFF-TA can be used to apply foundation models to high-resolution images without increasing compute. In certain scenarios, we find that training with LOFF-TA yields better results than directly fine-tuning the foundation model.

## 1 Introduction

Large and expensive foundation models, designed to capture general-purpose knowledge, have become a significant focus in machine learning and computer vision research [3]. These models excel in zero- and few-shot learning [4, 23] and adapt to various domains, especially in data-scarce scenarios, through transfer learning. But adapting these models for a specific task is a resource-intensive process [35]. The cost of fine-tuning large foundation models today is already prohibitive to most individuals and organizations. As they continue to grow in size, the rising costs of foundation models risk excluding all but the wealthiest organizations. To mitigate this, parameter-efficient fine-tuning methods have been proposed. These methods incorporate rank-deficient [18] and simple affine modules [31] or learnable prompt parameters injected at the input stage [20]. Their core principle is to limit the number of parameters that need to be trained. We extend this principle to its logical conclusion by introducing no intermediate parameters or prompts, and investigate whether it is possible to completely separate the foundation model from the training process.

In this study, we explore this complete separation of the resource-intensive foundation model from the training process. As seen in Figure 1, training data is passed through the foundation model at a one-time cost, and then cached. The cached feature embeddings are later loaded and used to train a lightweight classifier. By adopting this caching strategy we can train at a significantly faster rate using less memory resources and achieve similar,

38th Conference on Neural Information Processing Systems (NeurIPS 2024).

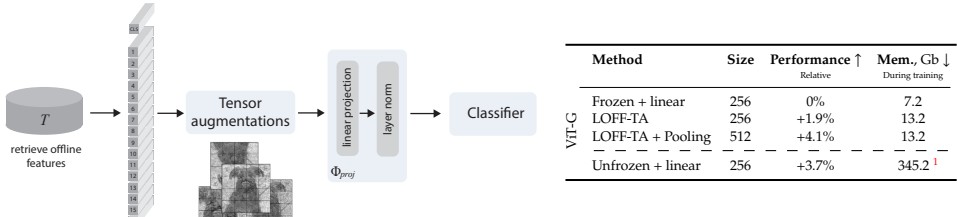

Figure 1: *Learning from Offline Foundation Features with Tensor Augmentations (LOFF-TA).* Training data is passed through a foundation model and cached. The cached embeddings are loaded and spatial tensor augmentations are applied in lieu of standard image augmentations. A lightweight classifier is trained on the cached, augmented features. This enables the use of arbitrarily large foundation models and high-resolution images at no additional cost.

or even in some cases, better performance. This framework enables us to leverage the power of *foundation models of any size* in limited resource settings. Moreover, tasks requiring high-resolution imaging, such as medical image diagnosis can benefit from the power of foundation models without increasing computational costs. However, these benefits come at the cost of inference speed, which can be slower as a consequence of our paradigm.

We call this approach "Learning from Offline Foundation Features with Tensor Augmentations", or LOFF-TA. It is a simple approach, but has not yet been explored as it is complementary to existing adaptation methods. LOFF-TA can be trivially combined with existing adaptation methods [18, 31, 20] by caching features from the adapted foundation models themselves to achieve even better performance. We conduct a series of comprehensive experiments using LOFF-TA on eleven well-known image classification benchmark datasets to demonstrate its benefits and limitations. Our key findings and contributions are outlined as follows:

- We propose, LOFF-TA which decouples the training process from the resource-intensive foundation model – a classifier is trained on cached features from foundation models instead of images.

- Since integrating image augmentations within LOFF-TA presents a challenge due to the tremendous storage cost of caching the embeddings of augmented images, we propose to apply spatial tensor augmentations to the cached embeddings of original images when training the compact classifier. We show that they perform nearly as well as standard image augmentations.

- We show that, using LOFF-TA, it is possible to achieve similar performance to a fine-tuned foundation model at a fraction of the computational cost – training speed is accelerated up to 37×, and GPU memory usage is reduced up to 26×. Additionally, LOFF-TA allows flexibility in choosing any input image size or foundation model according to need and available resources.

- Surprisingly, in some cases, LOFF-TA outperforms a fine-tuned foundation model.

Despite the simplicity of our approach, our findings indicate that there are many potential benefits, in terms of both performance and economics, to training from cached foundation features. The source code used in this work can be found at https://github.com/emirkonuk/loffta.

## 2 Related work

Foundation models [3] such as LLaMA [38] and GPT [4] have revolutionized natural language processing through large-scale training, requiring huge datasets and substantial computational resources. In a rapidly advancing competitive landscape, these models are

---

[1] It was not possible to train ViT-G on a single GPU with batch size of 64. Instead we report the memory footprint across 8 NVIDIA Quadro RTX 8000 using distributed training.

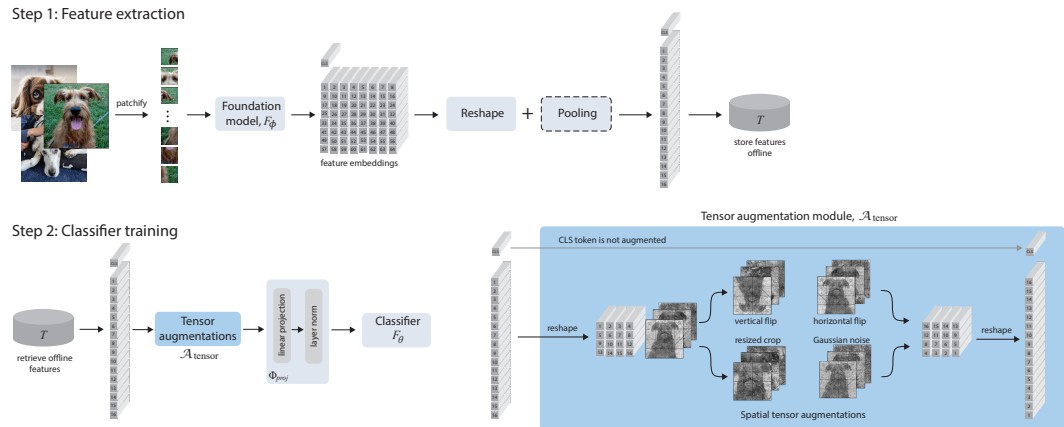

Figure 2: *Overview of LOFF-TA.* **Step 1:** We leverage a foundation model to process the training data and store the extracted features offline. **Step 2:** The cached tensors are loaded, tensor augmentations are applied, then the augmented tensors are passed through projection and normalization layers and used to train a lightweight classifier. The tensor augmentations include spatial-based transforms, such as flips and crops, along with additive Gaussian noise. An optional pooling step (dashed operation) reduces the spatial dimension of the stored features, allowing for training with high-resolution images at no additional cost.

scaling up dramatically to achieve unprecedented capabilities. In computer vision, this trend is mirrored with models like DINOv2 [35], CLIP [37], OpenCLIP [19], BLIP [30], SAM [23] and SEEM [47].

The standard approach for adapting a foundation model for a specific task involves fine-tuning its parameters or integrating additional trainable layers or classifiers [3, 13]. However, as the sizes of foundation models expand, these computational demands will escalate, posing significant challenges. To overcome these challenges, researchers have developed several strategies. These include fine-tuning only a subset of a model's parameters [46] and using techniques like gradient check-pointing for resource optimization [7]. Adaptation methods like [31, 17, 6, 18] introduce a limited number of tunable modules within the foundation model and incorporate learnable 'prompts' in the stem [20]. The result from these approaches is an efficiently fine-tuned, slightly bigger foundation model. Prior to deep learning, machine learning typically involved a two-stage process: first extract relevant features from data, then train a model on these features [2]. LOFF-TA, with its caching of features, echoes this latter approach [15]. As such, it complements the adaptation methods [31, 17, 6, 18, 20] and can be used in conjunction with them to improve overall performance, as we show in Section 5.5.

**Augmentations.** Developing novel image augmentation strategies mostly rely on visual insights to design augmentations [45]. The underlying principle is the manifold hypothesis [5] and effective augmentations should not move samples too far from the image manifold. While previous work on feature augmentations relied on interpolating features of different images [42] and adding noise to these interpolations [11] to ensure the augmented features stay close to the manifold, we find that feature augmentations need not be so limited. In this work, we show that it is surprisingly beneficial to apply spatial augmentations to features, in a manner similar to image augmentations.

## 3   Methods

In this work, we propose to pass training data through the foundation model, cache its features, and use them to train a lightweight classifier. We term this *Learning from Offline Foundation Features* (LOFF). A key challenge is LOFF's inability to incorporate image augmentations. To overcome this, we apply spatial and noise tensor augmentations directly

to the pre-stored foundation features, resulting in LOFF-TA. Finally, to allow the use of high-resolution images, we pool the foundation features.

## 3.1 LOFF

Our approach embraces a straightforward yet powerful idea: to separate feature extraction using a powerful foundation model from the training process using a lightweight classifier, as depicted in 2. We use a foundation model to process training data upfront, then extract and store the output features (Step 1). These serve as rich representations of the data which can be retrieved at a later time for training one (or more) classifiers, as depicted in Step 2.

In detail, given a dataset $D$, we generate and store a new dataset $T$ by applying a foundation model $F_\phi$ to all images $x$ from $D$. Each sample $t$ in $T$ is a $k \times d$ dimensional tensor where $k$ is the number of tokens for each input image $x$ and $d$ is the embedding dimensionality.

$$T := \{t \in \mathbb{R}^{k \times d}, \mid t = F_\phi(x)\}. \tag{1}$$

Importantly, *this phase only needs to be performed once*. The extracted features, $T$, along with the corresponding labels for each image, are cached and used as a replacement for $D$. The offline foundation features $T$ are retrieved to train a lightweight classification model $F_\theta$. Finally, we employ a standard cross-entropy loss, denoted as $\mathcal{L}$:

$$\arg\min_\theta \mathcal{L}(y, F_\theta(\Phi_{proj}(t))) \tag{2}$$

where, $y$ is the true label, $t \in T$ the stored features and $\Phi_{proj}$ is a projection module. The role of the projection module $\Phi_{proj}$ is to match the cached features to the dimensions expected by the classification model. $\Phi_{proj}$ consists of a learnable linear projection layer followed by a Layer Normalization (LN) operation [1], which we found to improve performance.

**High-resolution images** Many foundation models are trained to handle images larger than $224 \times 224$, such as [23, 35], but GPU memory limitations on conventional hardware make it impossible for them to be fine-tuned on high-resolution images. LOFF can mitigate this issue by pooling the features before storing them as shown in Fig. 2 (Step 1). The reduced size of the pooled features allow them to be stored and used to train a classifier efficiently. We investigate both average pooling and max pooling operations [27] and assess their impact on classification performance and computational cost.

## 3.2 LOFF-TA

Given that image augmentations are crucial for effective model training, the inability to apply them in the LOFF framework poses a significant challenge. The obstacle lies in the impracticality of storing tensors from augmented images during Step 1, which would result in prohibitive storage demands. To address this drawback, we introduce tensor augmentations $\mathcal{A}_{\text{tensor}}(t)$ on the features $t \in T$. A tensor augmentation module is applied dynamically online, before a batch of features is fed into the classifier, as depicted in 2. This method, named *Learning from Offline Foundation Features with Tensor Augmentations* (LOFF-TA), allows augmentations to be used for overcoming the aforementioned obstacle with image augmentations.

In a standard setting, the objective for image classification is given by

$$\arg\min_\theta \mathcal{L}(y, F_\theta(\mathcal{A}_{\text{img}}(x))) \tag{3}$$

where $F_\theta$ is the model to be trained on each sample $(x, y)$ from the dataset $\mathcal{D}$ and $\mathcal{A}_{\text{img}}$ indicates stochastic image augmentations. LOFF-TA changes the optimization task to become

$$\arg\min_\theta \mathcal{L}(y, F_\theta(\Phi_{proj}(\mathcal{A}_{\text{tensor}}(t)))) \tag{4}$$

where $\mathcal{A}_{\text{tensor}}$ denotes our tensor augmentation operator. The stored features $t \in T$ and projection module $\Phi_{proj}$ remain the same as in LOFF.

Table 1: *Main results.* We train models on features extracted by DINOv2 [35] ViT-B and ViT-G models. We report the results using LOFF (no augmentations) and LOFF-TA (with tensor augmentations). We consider features extracted from $256 \times 256$ and $512 \times 512$ images (using pooling as described in 3.1). *Frozen + linear* and *Unfrozen + linear* are points of comparison consisting of a frozen/unfrozen foundation model with a linear layer trained on images directly, with image augmentations.

| | Method | Size | APTOS, $\kappa$ ↑ $n = 3{,}662$ | AID, Acc. ↑ $n = 10{,}000$ | DDSM, AUC ↑ $n = 10{,}239$ | ISIC, Rec. ↑ $n = 25{,}333$ | NABirds, Acc. ↑ $n = 48{,}562$ | TP, Im/sec ↑ Train (Infer.) | Mem.,Gb ↓ Training |
|---|---|---|---|---|---|---|---|---|---|
| ViT-B | Frozen + linear | 256 | 88.6 ± 0.3. | 90.9 ± 0.1 | 90.3 ± 0.2 | 51.7 ± 1.0 | 86.0 ± 0.1 | 153 (313) | **1.8** |
| | LOFF | 256 | 89.6 ± 0.2 | 91.9 ± 0.3 | 94.2 ± 1.2 | 70.8 ± 2.1 | 83.0 ± 0.1 | **228** (236) | 13.2 |
| | LOFF-TA | | 90.4 ± 0.6 | 92.3 ± 0.7 | 94.4 ± 0.1 | 72.8 ± 1.7 | 83.5 ± 0.3 | 227 (236) | 13.2 |
| | LOFF + Pool | 512 | 89.4 ± 1.5. | 93.2 ± 0.6 | 95.3 ± 0.5 | 74.3 ± 1.5 | 86.2 ± 0.3 | **228** (61) | 13.2 |
| | LOFF-TA + Pool | | **90.5 ± 1.0** | **93.7 ± 0.3** | **95.5 ± 0.1** | **77.4 ± 0.0** | **86.8 ± 0.4** | 227 (61) | 13.2 |
| | Unfrozen + linear | 256 | 90.5 ± 0.9 | 93.7 ± 0.8 | 93.3 ± 0.9 | 76.8 ± 0.7 | 85.8 ± 0.1 | 77 (313) | 28.2 |
| ViT-G | Frozen + linear | 256 | 88.2 ± 0.3 | 92.8 ± 0.2 | 90.8 ± 0.6 | 66.4 ± 1.1 | 89.8 ± 0.2 | 14 (28) | **7.2** |
| | LOFF | 256 | 88.6 ± 1.5 | 93.3 ± 0.5 | 94.8 ± 1.6 | 73.1 ± 0.5 | 87.4 ± 0.2 | **222** (27) | 13.2 |
| | LOFF-TA | | 89.9 ± 0.4 | 94.0 ± 0.2 | 95.3 ± 0.1 | 76.0 ± 0.7 | 88.5 ± 0.2 | 218 (27) | 13.2 |
| | LOFF + Pool | 512 | 90.3 ± 0.6 | 94.1 ± 0.2 | 95.4 ± 0.4 | 74.0 ± 1.6 | 88.8 ± 0.1 | **222** (7) | 13.2 |
| | LOFF-TA + Pool | | **91.8 ± 0.3** | **94.6 ± 0.2** | **96.3 ± 0.6** | **79.9 ± 0.2** | **90.1 ± 0.2** | 218 (7) | 13.2 |
| | Unfrozen + linear | 256 | 89.6 ± 0.6 | 96.2 ± 0.1 | 96.7 ± 0.2 | 87.3 ± 1.3 | 90.2 ± 0.1 | 6 (28) | 345.2 [1] |

## 3.3 Tensor augmentations

Spatial relationships in image data are crucial for understanding the content of the image. Directly applying augmentations haphazardly to the unstructured output tokens from the foundation model may lead to undesirable results. A key aspect of our approach involves the utilization of spatial tensor augmentations during the training phase, as depicted in 2. These augmentations are chosen to consider the spatial relationships in the data, similar to how image augmentations operate. We apply spatial augmentations, denoted as $\mathcal{A}_{\text{tensor}}$, to foundation features $t \in T$ after a reshaping operation. These augmentations are conceptually analogous to image augmentations, treating the foundation features as if they were low-resolution, hyper-spectral images. We select a set of *spatial augmentations* suited for feature-level transformations, chosen to enhance the training process while maintaining the integrity of spatial relationships. We *flip* by mirroring the tensor on its height or width axis, *resize* it by upsampling or downsampling its spatial dimensions using linear interpolation. We *shear* the tensor using nearest neighbor interpolation and *translate* it by shifting along its spatial dimensions. We *rotate* the tensor in its spatial dimensions around its spatial center using nearest neighbor interpolation. In addition to spatial augmentations, we apply additive *Gaussian noise* with zero mean to the feature tensor, similar to [11]. Although we considered channel augmentations, analogous to contrast or color augmentations in images, we did not find them to be beneficial. We also note that, like image augmentations, not all tensor augmentations types are appropriate in every setting[2].

## 4 Experimental setup

To evaluate the effectiveness of LOFF-TA we benchmark over eleven datasets from various domains using different foundation models, model capacities and image resolutions.

### 4.1 Models and implementation details

**Foundation models.** We employ two foundation model families: DINOv2 [35] and CLIP [37] (implemented by OpenCLIP [19]) as the basis for our investigations. For the majority of our experiments, we utilize the ViT-B and ViT-G architectures.

**Classifiers.** LOFF and LOFF-TA train a lightweight classifier on the features from a foundation model. While, in principle, any classifier can be used in this role, our experiments use DeiT-S [39]. The classifier is initialized using ImageNet [10] pre-trained weights, as we found empirically this gave a significant improvement over random initialization [14]. In some

---

[2]We omit vertical flips for the SUN397 dataset [44].

Table 2: *Expanded results on seven standard datasets.* We compare LOFF (no augmentations) and LOFF-TA (with tensor augmentations) against baselines *Frozen + linear*, *Unfrozen + linear* and *Frozen + DeiT-S* consisting of a frozen/unfrozen foundation model with a linear layer/DeiT-S classifier trained on images directly (with image augmentations). Results are reported for features extracted from OpenCLIP [19] and DINOv2 [35] using 256×256 images.

| | | Method | Oxford-III Pet $n = 7,349$ | Flowers102 $n = 8,189$ | Caltech-101 $n = 8,677$ | StanfordCars $n = 16,185$ | StanfordDogs $n = 20,580$ | SUN397 $n = 39,700$ | NABirds $n = 48,562$ | TP / Mem. Im/s ↑ / Gb ↓ |
|---|---|---|---|---|---|---|---|---|---|---|
| DINOv2 | ViT-B | Frozen + linear | 95.7 ± 0.1 | 99.7 ± 0.1 | 96.7 ± 0.4 | 87.9 ± 0.1 | 87.8 ± 0.1 | 75.4 ± 1.2 | 86.0 ± 0.1 | 153 / 1.8 |
| | | LOFF | 94.5 ± 0.4 | 99.2 ± 0.1 | 96.2 ± 0.7 | 87.7 ± 0.6 | 84.6 ± 0.3 | 76.5 ± 0.1 | 83.0 ± 0.1 | 228 / 13.2 |
| | | LOFF-TA | 95.2 ± 0.1 | 99.5 ± 0.1 | 97.0 ± 0.5 | 88.9 ± 0.4 | 85.3 ± 0.2 | 76.8 ± 0.1 | 83.5 ± 0.3 | 227 / 13.2 |
| | | Unfrozen + linear | 94.8 ± 0.2 | 99.0 ± 0.1 | 97.1 ± 0.3 | 93.7 ± 0.1 | 86.4 ± 0.4 | 76.2 ± 0.2 | 85.8 ± 0.1 | 77 / 28.2 |
| | | Frozen + DeiT-S | 94.8 ± 1.0 | 99.6 ± 0.0 | 97.0 ± 0.3 | 91.6 ± 0.2 | 87.4 ± 0.4 | 76.8 ± 0.1 | 85.4 ± 0.1 | 94 / 13.8 |
| DINOv2 | ViT-G | Frozen + linear | 96.2 ± 0.1 | 99.7 ± 0.0 | 96.1 ± 0.4 | 90.2 ± 0.2 | 89.8 ± 0.1 | 78.2 ± 0.1 | 89.8 ± 0.2 | 14 / 7.2 |
| | | LOFF | 95.5 ± 0.2 | 99.7 ± 0.1 | 96.0 ± 0.5 | 91.8 ± 0.0 | 89.0 ± 0.3 | 78.2 ± 0.2 | 87.4 ± 0.2 | 222 / 13.2 |
| | | LOFF-TA | 95.8 ± 0.4 | 99.7 ± 0.1 | 96.8 ± 0.3 | 92.8 ± 0.1 | 89.2 ± 0.1 | 79.2 ± 0.3 | 88.5 ± 0.2 | 218 / 13.2 |
| | | Unfrozen + linear | 96.0 ± 0.2 | 99.7 ± 0.1 | 97.5 ± 0.1 | 94.5 ± 0.6 | 90.1 ± 0.2 | 79.6 ± 0.6 | 90.2 ± 0.1 | 6 / 345.2 [1] |
| | | Frozen + DeiT-S | 96.1 ± 0.1 | 99.7 ± 0.0 | 96.7 ± 0.2 | 93.3 ± 0.3 | 89.3 ± 0.1 | 78.7 ± 0.3 | 88.9 ± 0.2 | 13 / 18.2 |
| OpenCLIP | ViT-B | Frozen + linear | 91.7 ± 0.3 | 90.4 ± 0.6 | 96.0 ± 0.3 | 94.1 ± 0.3 | 76.8 ± 0.1 | 77.7 ± 0.0 | 61.0 ± 0.3 | 206 / 1.8 |
| | | LOFF | 91.4 ± 0.3 | 89.2 ± 0.1 | 94.6 ± 0.8 | 93.3 ± 0.2 | 72.7 ± 1.2 | 77.5 ± 0.1 | 70.7 ± 0.1 | 228 / 13.2 |
| | | LOFF-TA | 91.8 ± 0.2 | 94.1 ± 0.7 | 95.4 ± 0.1 | 93.4 ± 0.1 | 74.1 ± 0.9 | 77.7 ± 0.2 | 71.2 ± 0.2 | 227 / 13.2 |
| | | Unfrozen + linear | 92.5 ± 0.2 | 96.4 ± 0.2 | 96.4 ± 0.5 | 94.2 ± 0.2 | 80.1 ± 0.7 | 75.7 ± 0.1 | 79.1 ± 0.1 | 101 / 12.9 |
| | | Frozen + DeiT-S | 92.9 ± 0.2 | 95.3 ± 0.3 | 96.0 ± 0.1 | 94.4 ± 0.4 | 79.5 ± 0.4 | 78.1 ± 0.3 | 72.1 ± 0.2 | 124 / 14.5 |

experiments, we compare against a frozen/unfrozen foundation model as a benchmark, with either a linear layer or DeiT-S on top as a classifier.

**Implementation details.** In our experiments, we utilize the AdamW optimizer [32], and a batch size of 64. We incorporate a learning rate warm-up strategy and manually decrease the learning rate by a factor of 0.1 when the validation performance plateaus. For lightweight classifier in LOFF and LOFF-TA, we implement modifications to the DeiT-S architecture [39]. We remove the patchifier from the model's stem and introduce a linear projection layer followed by a normalization layer as detailed in 3.

## 4.2 Datasets

Our evaluation spans eleven image classification datasets, covering a diverse spectrum of object categories. We begin with datasets with high-resolution images, as this setting highlights new capabilities made possible by LOFF-TA. We include APTOS2019 [21] for diabetic retinopathy detection, DDSM [29] for identifying masses in mammography, ISIC [40, 8, 9] for skin lesion classification, AID [43] for aerial image classification, and NABirds [41] for fine-grained bird species classification. The resolution of these datasets varies, but we resize them to 512 × 512. We extend our evaluation to a number of standard 256 × 256 resolution benchmark datasets: Flowers102 [34], NABirds [41], StanfordCars [26], StanfordDogs [22], Oxford-III Pet [36], Caltech-101 [12], and SUN397 [44]. For each dataset, we report metrics appropriate to its specific evaluation criteria. We adhere to official train/validation/test splits when available, or follow [24] in their absence. Standard practice of image normalization is maintained, using the mean and variance from the training sets.

## 5 Results

In this section we show that LOFF-TA achieves competitive, sometimes superior, results compared to the baselines while significantly reducing memory usage and training time.

### 5.1 Effectiveness of LOFF-TA

We begin our analysis with Table 1 where we focus on the cost-benefit trade-off of using LOFF-TA, considering DINOv2 as the foundation model. Our evaluation spans five datasets where higher resolution images are known to improve performance. We measure each approach in terms of performance, throughput (TP), and memory (Mem.) footprint. The study includes a comparison with two key configurations: a baseline approach, where a linear classifier is appended to a frozen foundation model (*frozen + linear*), and an upper-bound where the entire foundation model is fine-tuned in a typical fashion for transfer

Table 3: *Ablations.* We systematically remove components of LOFF-TA to investigate the impact of each contribution. Results are reported with DINOv2 [35] as the foundation model, adding trivial augment [33] as an augmentation strategy.

| Foundation CLS | Layer norm | Gaussian noise | Spatial aug. | Trivial augment [33] | Oxford-III Pet | Caltech-101 |
|:---:|:---:|:---:|:---:|:---:|:---:|:---:|
| ✗ | ✓ | ✓ | ✓ | ✗ | 94.1 | 95.6 |
| ✓ | ✗ | ✓ | ✓ | ✗ | 94.9 | 94.8 |
| ✓ | ✓ | ✗ | ✗ | ✗ | 94.9 | 95.4 |
| ✓ | ✓ | ✗ | ✓ | ✗ | 95.1 | 96.2 |
| ✓ | ✓ | ✓ | ✓ | ✗ | 95.2 | 96.4 |
| ✓ | ✓ | ✗ | ✗ | ✓ | **95.4** | **96.8** |

learning (*unfrozen + linear*). Both employ standard image augmentations. In contrast, LOFF operates without augmentations, while LOFF-TA introduces tensor augmentations. We also explore the effects of pooling on foundational features, increasing image resolution from 256 to 512, to understand how these adjustments influence the performance and efficiency.

Our observations reveal several trends. Firstly, all LOFF and LOFF-TA variants (with an exception of NABirds) surpass the baseline in performance, with only a slight increase in memory usage but significantly faster training. Secondly, LOFF-TA consistently outperforms LOFF, confirming the importance of tensor augmentations. Moreover, upgrading to a larger foundation model (from ViT-B to ViT-G) doesn't alter memory or training speed. Pooling and working with higher resolution further improves performance, again without affecting memory or throughput. Remarkably, in many cases (6 out of 10), LOFF-TA with pooling exceeds the performance of the intended upper-bound model (*unfrozen + linear*). Finally, it's worth noting the training speed and memory efficiency of LOFF and LOFF-TA compared to fine-tuning the foundation model (*unfrozen + linear*); our approach offers a 37× acceleration during training, and a remarkable savings in memory as well – in fact, ViT-G is too large to fine-tune on a conventional GPU with a batch size of 64.

## 5.2 Further evidence

In Table 2, we continue our analysis in a similar fashion to Table 1, but in this case focusing on seven standard visual object recognition datasets. Our focus remains on assessing performance, throughput (TP), and memory usage (Mem.), but with image resolution of 256. The analysis compares LOFF, LOFF-TA, *frozen + linear*, and *unfrozen + linear*, and introduces *frozen + DeiT-S* as a benchmark for image vs. tensor augmentations (see Section 5.3).

The findings in Table 2 reinforce the patterns observed in Table 1. LOFF-TA maintains its superiority over LOFF, highlighting the efficacy of tensor augmentations. It's important to note that in this set of experiments, the best performing LOFF-TA + Pooling configuration using high-resolution images is not considered due to the datasets' inherent resolution limit of 256. Despite this, LOFF-TA generally matches or surpasses the baseline *frozen + linear* in performance, while delivering a notable increase in throughput. Although *frozen + linear* claims the smallest memory footprint, LOFF and LOFF-TA again show significant savings compared to fine-tuning a foundation model (*unfrozen + linear*). Comparing foundation models of similar capacity, we notice that DINOv2 outperforms OpenCLIP in five of the seven datasets using LOFF-TA. Finally, we once again see that swapping the foundation model (between DINOv2 ViT-B, OpenCLIP ViT-B, and DINOv2 ViT-G) results in no appreciable change in throughput or memory consumption for LOFF-TA, only differences in performance.

## 5.3 Image vs. tensor augmentations

Looking at Table 2 again, we investigate the effect of image augmentations versus tensor augmentations. LOFF-TA and *frozen + DeiT-S* share the same architecture (a cascaded model that consists of a frozen foundation model with an appended DeiT-S), and in both cases the foundation is frozen – the only difference is LOFF-TA uses tensor augmentations while *frozen + DeiT-S* uses image augmentations. We also report results (Table 5 in the Appendix) when we unfreeze the foundation model in the cascaded setting for both OpenCLIP and DINOv2 for completeness. In Table 2, we observe that image augmentations outperform

tensor augmentations. As one might expect, tensor augmentations such as tensor rotation or cropping can not replicate the exact effects of image rotation or cropping since foundation models are not linear operators. Yet, the performance impact is surprisingly less than anticipated, which, considering LOFF-TA's significant computational savings, is noteworthy.

## 5.4 Ablation study

In this section, we try to understand the impact of each contribution to LOFF-TA. Utilizing DINOv2 [35], we conduct a study where components are systematically removed and then tested on the Oxford-III Pet and Caltech-101 datasets. Results are presented in Table 3.

**CLS token.** A perhaps obvious, but critical, insight from our study is the important role played by the foundation model's CLS token in contributing to classifier performance. When this token, representing global information, is integrated into the classifier training, it proves to be a key contributor to performance gains. However, the choice of how to integrate it was not trivial. We opted to integrate the offline CLS token from the foundation model with the learned CLS token of the classifier model by summation. Other possible ways to incorporate it could be to initialize the classifier's CLS to the offline foundation CLS, or concatenate them – although our non-exhaustive experiments testing these approaches were inferior.

**Layer norm.** We applied a Layer Norm [1] operation in the projection layer at the stem of the classifier. Similar to the CLS token ablation, we find that adding a layer norm operation before passing the foundation features to the classifier boosts performance. By centering input features, we believe the normalization plays a pivotal role similar to that in Dual Patchnorm [28], aligning and standardizing the input.

**Pooling.** Table 1 shows that pooling larger image features enhances performance without extra computational costs, enabling the use of large models for high-resolution tasks. In Appendix Table 6, we compare max and average pooling, finding both perform similarly, with max pooling slightly outperforming in larger models, possibly due to better noise reduction. Thus, max pooling is recommended for larger models. While this work focuses on pooling, alternative dimensionality reduction methods like strided convolutions or bi-linear interpolation could also be effective.

**Augmentations.** Since the cached features contain long-range spatial information that may potentially be harmed by our tensor augmentations, it is important to assess if these augmentations have a meaningful contribution to the performance of LOFF-TA. Comparing Row 3 and Row 4 of Table 3, we see that adding spatial augmentations results in a significant boost in performance. A smaller boost is observed when Gaussian noise is added in Row 5. Replacing our augmentations with Trivial augment [33] provides a further boost to performance, although this setup was not used in our other experiments.

## 5.5 LOFF-TA vs. foundation adaptation methods

LOFF-TA enables practitioners to use large foundation models *without any modifications*. A class of methods exists that modify foundation models for new tasks, so-called adaptation methods. Adaptation methods can be used in conjunction with LOFF-TA, but we also provide a comparison between them in Table 4. The results show that standalone LOFF-TA achieves competitive performance with VPT [20], Adaptformer [6] and SSF [31]. When LOFF-TA is combined with these methods, we observe noticeable improvements across the board, demonstrating that computationally efficient SOTA performance can be achieved by adapting foundation models and then applying LOFF-TA.

## 6 Discussion

**Performance-efficiency trade-off.** The aim of this paper is to consider the use of foundation models in a resource-limited setting. To do so, we propose to work with cached foundational features. The benefits of doing so are that training speed is significantly accelerated (up to 37×), and GPU memory usage is significantly reduced (up to 26×). Of course, these benefits

Table 4: *LOFF-TA can be used alongside foundation adaptation methods*. Below we report results using ViT-B. Standalone LOFF-TA performs comparable with other foundation adaptation methods and can easily be combined with them to further enhance performance.

| Method | APTOS, $\kappa$ ↑ $n = 3{,}662$ | AID, Acc. ↑ $n = 10{,}000$ | DDSM, AUC ↑ $n = 10{,}239$ | ISIC, Rec. ↑ $n = 25{,}333$ | NABirds, Acc. ↑ $n = 48{,}562$ |
|---|---|---|---|---|---|
| LOFF-TA | 90.4 ± 0.6 | 92.3 ± 0.7 | 94.4 ± 0.1 | 72.8 ± 1.7 | 83.5 ± 0.3 |
| VPT [20] | 89.6 ± 0.1 | 93.0 ± 0.1 | 91.4 ± 0.3 | 75.2 ± 1.1 | 85.8 ± 0.2 |
| VPT + LOFF-TA | 90.8 ± 0.4 | 93.1 ± 0.3 | 92.4 ± 0.3 | 79.7 ± 0.9 | 83.7 ± 0.1 |
| SSF [31] | 90.2 ± 0.1 | 92.1 ± 0.2 | 96.7 ± 0.6 | 76.4 ± 0.9 | 88.2 ± 0.0 |
| SSF + LOFF-TA | 91.1 ± 0.7 | 93.1 ± 0.0 | 97.2 ± 0.3 | 81.6 ± 1.5 | 85.6 ± 0.1 |
| AdaptFormer [6] | 89.6 ± 0.6 | 94.3 ± 0.1 | 91.8 ± 0.8 | 82.6 ± 1.0 | 87.1 ± 0.3 |
| AdaptFormer + LOFF-TA | 90.0 ± 0.3 | 94.3 ± 0.2 | 93.2 ± 0.5 | 83.5 ± 0.3 | 85.3 ± 0.2 |

come with trade-offs. In most cases, LOFF-TA will perform slightly worse than directly fine-tuning the foundation model, although we did observe some cases where LOFF-TA was superior. Also, the computational cost savings at training time do not translate to inference.

An important benefit of LOFF-TA is that it affords flexibility, allowing practitioners to choose the most suitable foundation model for their task from a range of sizes and pretraining methods (refer to Table 2), including the capacity to handle high-resolution images (see Table 1). The caching and pooling of features makes these different choices come with equivalent computational costs during training, determined by the compact classifier.

**Societal impact:** As foundation models grow, their computational demands are expected to increase, potentially creating a digital divide where advanced models are accessible only to well-resourced organizations, excluding smaller entities and individual researchers. This trend underscores the importance of developing efficient methods to utilize these models, ensuring broad access and fostering progress in various application areas. There is significant potential to impact fields like medical imaging and remote sensing, which can greatly benefit developing regions and marginalized groups. But these applications often require high-resolution image processing, which is resource-intensive. LOFF-TA facilitates wider, fairer access to sophisticated foundation models in resource-limited settings.

**Limitations:** LOFF-TA is a training strategy meant for low resource settings. Given enough computational resources, full fine-tuning will generally outperform LOFF-TA or adaptation methods like SSF [31]. Another limitation of LOFF-TA is that at inference time it is less efficient than standalone foundation models. However, we note that due to their latency, foundation models are costly to be deployed at scale without distillation [16] (or some other measure) to limit cost which would also alleviate the limitation for LOFF-TA.

## 7  Conclusion

Foundation models have significantly impacted the community and will likely become more crucial as they grow in size and capability. However, adapting these models for specific tasks is increasingly challenging due to their high resource demands. Revisiting the classical machine learning paradigm of *perceive, then reason* is pragmatic in this context. Using foundation models as powerful feature extractors allows us to benefit from their rich representations, while at the same time mitigating computational costs by training a lightweight classifier on their cached features. LOFF-TA achieves a low memory footprint and high throughput during training, regardless of the chosen foundation model or input image size. It is particularly beneficial for high-resolution image processing when pooling is applied, allowing LOFF-TA to outperform linear classifiers using frozen foundation models and compete with or surpasses fine-tuned foundation models.

**Acknowledgements.** This work was supported by the Wallenberg AI, Autonomous Systems and Software Program (WASP) and the Development and Promotion of Science and Technology Talents Project. We acknowledge the Berzelius computational resources provided by the Knut and Alice Wallenberg Foundation at the National Supercomputer Centre.

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

# Appendix

## A Representation similarities

To examine the similarities between classifiers trained on images and trained on tensors using LOFF, we employed Centered Kernel Alignment (CKA) [25], a technique commonly used to analyze representational similarity in neural networks. Figure 3 presents the CKA analysis results for classifiers trained on Oxford-III Pet. In the top panels, we observe that the LOFF classifier's low-to-mid layers exhibit similarity to the higher layers of the image-trained classifier, indicating that LOFF learns features similar to the image-trained classifier's high-level features. This suggests that LOFF's offline features are sufficiently informative for learning high-level features earlier in the network, with subsequent layers adapting to novel features. The middle panels demonstrate that the cascaded frozen foundation model classifier displays similar behavior to the LOFF classifier. The right panel reveals high similarities between corresponding layers of the frozen foundation and LOFF classifiers, particularly in the earlier layers, indicating shared learned features. The bottom panel illustrates the internal representational similarity of each classifier with itself before and after fine-tuning. The left panel shows strong similarity between the image-trained classifier's layers, indicating the retention of pretrained features during fine-tuning. Conversely, the middle and right panels indicate that the frozen foundation and LOFF classifiers retain similarity in the low-to-mid layers while learning new features in the higher layers.

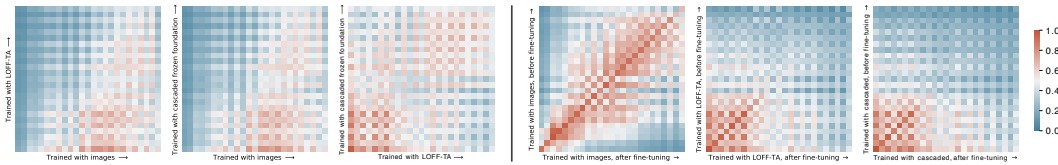

Figure 3: *CKA similarities between different models. Left*: Representation similarity of different classifiers after fine-tuning on Oxford-III-Pet. *Right*: Representation similarity of the internal layers of each classifier with itself before and after fine-tuning.

## B Unfreezing the Cascaded Models

We evaluate the model's performance in the cascaded setting (*foundation model + DEIT-S classifier*) when the whole model is fine-tuned. We present our findings in Table 5, comparing the use of DINOv2 and OPENCLIP as foundation models. Once again, we find that DINOv2 outperforms OPENCLIP, in most cases. When DINOv2 is unfrozen, it either surpasses or matches the performance of its frozen alternative (see Table 2), though the margin of improvement is rather small. OPENCLIP shows a similar trend, but its performance seems more dependent on the dataset it is evaluated on. Overall, unfrozen foundation models appear to outperform their frozen alternatives. However, their computational cost makes them prohibitive for larger models or image sizes.

Table 5: *Fine-tuning a foundation model + DEIT-S classifier*. We append a DEIT-S classifier after a DINOv2 [35] or a OPENCLIP [19] foundation model and the whole cascaded model.

| Foundation | Oxford-III Pet $n = 7{,}349$ | Flowers102 $n = 8{,}189$ | Caltech-101 $n = 8{,}677$ | StanfordCars $n = 16{,}185$ | StanfordDogs $n = 20{,}580$ | SUN397 $n = 39{,}700$ | NABirds $n = 48{,}562$ |
|---|---|---|---|---|---|---|---|
| DINOv2 ViT-B | 95.2 ± 0.2 | 99.3 ± 0.1 | 97.4 ± 0.2 | 93.9 ± 0.4 | 87.0 ± 0.3 | 76.2 ± 0.7 | 86.3 ± 0.2 |
| OPENCLIP ViT-B | 91.8 ± 0.5 | 95.4 ± 1.2 | 96.8 ± 0.2 | 94.5 ± 0.2 | 81.3 ± 0.3 | 74.8 ± 0.2 | 77.8 ± 0.1 |

## C   Tensor augmentations

In our experiments, we consistently observe a performance improvement when applying tensor augmentations. However, the need for task-specific and domain-appropriate augmentation strategies should be emphasized. For example, in scene classification using the SUN397 dataset, applying vertical flips to images when training DɛIT-S (pretrained on ImageNet) led to decreased accuracy (−2%), likely due to the unrealistic expectation of upside-down building facades during testing. One might expect a similar effect with tensor augmentations, since the feature space preserves the spatial orientations of objects (see Figure 4). Interestingly we see a much smaller performance drop when we apply vertical flips as tensor augmentations on this dataset (−0.2%). This suggests LOFF-TA's tensor augmentations exhibit greater resilience to *improper* augmentations compared to the image domain. This observation merits further exploration to understand its underlying mechanisms and consequences.

Intriguingly, LOFF-TA demonstrates benefits from spatial tensor augmentations despite their potential conflict with the spatial information present in feature tokens. Feature tokens in foundation models carry positional data, informed by positional embeddings and attention mechanisms. Spatial augmentations, such as horizontal flips, disrupt this positional context, yet our experiments, especially with Trivial Augment, show a notable performance enhancement (see Table 3). This paradox suggests that these disruptions may actually bolster the classifier's ability to learn robust features, thereby improving classification outcomes. The precise dynamics of this phenomenon remain unclear, presenting an exciting avenue for future research, particularly in the realm of auto augment strategies for foundation features.

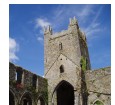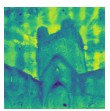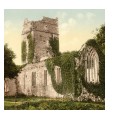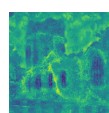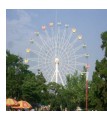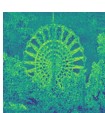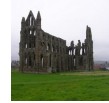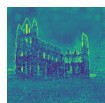

Figure 4: *Robustness and spatial consistency of features.* Images along with a random channel of the corresponding foundation features reveal the spatial consistency between objects in the image and feature spaces. This consistency allows insights from the image space to guide tensor augmentation choice, *e.g.* if vertical flips are harmful for a building facade dataset in image space, they are likely to be harmful in feature space. However, we observe that training with LOFF-TA is more robust against 'incorrect' augmentation choices compared to standard classifier training on images.

## D   Pooling ablation

Table 6: *Pooling enables larger resolution.* We compare different approaches to pooling used for LOFF and LOFF-TA: *no pooling*, *max pooling*, and *average pooling*. Results are reported for foundation features extracted by DINOv2 [35] ViT-B and ViT-G models from standard 256×256 without pooling versus larger 512×512 images with pooling (note that the compute costs are equivalent).

| | Method | Pooling | Size | APTOS, $\kappa$ ↑ $n = 3,662$ | AID, Acc. ↑ $n = 10,000$ | DDSM, AUC ↑ $n = 10,239$ | ISIC, Rec. ↑ $n = 25,333$ | NABirds, Acc. ↑ $n = 48,562$ |
|---|---|---|---|---|---|---|---|---|
| ViT-B | LOFF | ✗ | 256 | 89.6 ± 0.2 | 91.9 ± 0.3 | 94.2 ± 1.2 | 70.8 ± 2.1 | 83.0 ± 0.1 |
| | LOFF-TA | | | 90.4 ± 0.6 | 92.3 ± 0.7 | 94.4 ± 0.1 | 72.8 ± 1.7 | 83.5 ± 0.3 |
| | LOFF | Average | 512 | 90.3 ± 0.2 | 92.7 ± 0.7 | 95.7 ± 0.3 | 73.8 ± 0.1 | 86.3 ± 0.3 |
| | LOFF-TA | | | **90.7 ± 0.8** | **93.7 ± 0.5** | **96.1 ± 0.1** | **77.9 ± 1.9** | 86.7 ± 0.3 |
| | LOFF | Max | 512 | 89.4 ± 1.5. | 93.2 ± 0.6 | 95.3 ± 0.5 | 74.3 ± 1.5 | 86.2 ± 0.3 |
| | LOFF-TA | | | 90.5 ± 1.0 | **93.7 ± 0.3** | 95.5 ± 0.1 | 77.4 ± 0.0 | **86.8 ± 0.4** |
| ViT-G | LOFF | ✗ | 256 | 88.6 ± 1.5 | 93.3 ± 0.5 | 94.8 ± 1.6 | 73.1 ± 0.5 | 87.4 ± 0.2 |
| | LOFF-TA | | | 89.9 ± 0.4 | 94.0 ± 0.2 | 95.3 ± 0.1 | 76.0 ± 0.7 | 88.5 ± 0.2 |
| | LOFF | Average | 512 | 89.0 ± 0.6 | 94.0 ± 0.3 | 96.0 ± 0.5 | 77.5 ± 0.7 | 89.0 ± 0.2 |
| | LOFF-TA | | | 90.1 ± 0.7 | 94.3 ± 0.2 | 96.1 ± 0.6 | 79.4 ± 2.8 | 90.0 ± 0.2 |
| | LOFF | Max | 512 | 90.3 ± 0.6 | 94.1 ± 0.2 | 95.4 ± 0.4 | 74.0 ± 1.6 | 88.8 ± 0.1 |
| | LOFF-TA | | | **91.8 ± 0.3** | **94.6 ± 0.2** | **96.3 ± 0.6** | **79.9 ± 0.2** | **90.1 ± 0.2** |

