# OpenReview forum: "Learning from Offline Foundation Features with Tensor Augmentations"
_NeurIPS.cc/2024/Conference — NeurIPS 2024 poster_

### Official Review · Reviewer_AUq9 · 2024-07-10

**Soundness:** 2
**Presentation:** 2
**Contribution:** 2
**Rating:** 4
**Confidence:** 4

**Summary:**

The paper discusses leveraging the large foundational models in resource-scare settings to fine-tune an image classification task. The authors propose saving the features from the frozen foundational model and then using them to train/fine-tune a smaller classifier model. They introduce tensor augmentations to improve the method's robustness and demonstrate the advantages of reduction in GPU memory usage and improved training speed and performance on different open-source dataset tasks. The performance of the proposed method is comparable to the full fine-tuning of the foundations model along with a linear layer.

**Strengths:**

- This work is essential in low-resource settings, utilizing a much better backbone foundational model to improve classification accuracy.

- Results show improvement over the baseline methods and simple fine-tuning as well.

**Weaknesses:**

- Table 1 results. Question about the inference throughput. The table shows a considerable reduction compared to the original model with linear. I assume that inference is done end-to-end, i.e., the image passed through the Foundational Model + the extra bits of the proposed LOFF-TA/LOFF pipeline. While the improved training speed is appreciable, the drop in inference is much less desirable, especially when deployed in scenarios that require faster throughput.  So although the resource saving is highly beneficial for the training scenarios, unfortunately for inference the whole system still needs to be loaded onto the GPU because caching is not possible. I would appreciate the author's insight on this.

- The effectiveness of these augmentations depends heavily on the type of features extracted from the foundation model. Specifically, different layers within a neural network capture varying level of information: earlier layers focus on the spatial structure of the image, while deeper layers learn more abstract and semantic features. The paper needs to clarify what layer of the ViT features were extracted and cached. Further, there is a strong dependency on this choice of layers, and I don't see any ablations on this choice.

- It's interesting that pooling actually improves the results. It's surprising because I would assume pooling reduces some of the features when reducing the dimension, but it somehow actually helps. Can the authors comment on this? It's a little counterintuitive, in my opinion.

- Authors propose a $\phi_{proj}$ adapter to match the cached features to the dimensions expected in the classifier model. Is the layer trained end-to-end along with the classifier?

**Questions:**

See above

**Limitations:**

There is a lack of clarity in some of the choices in the design. I have a few questions on some of the results and the missing ablation experiments.

---

> ### Author Rebuttal · Authors · 2024-08-06
>
> We thank the reviewer for their comments.
>
> 1 - This is the main limitation of our study that we highlighted in the Limitations subsection - much like any caching strategy, there is a trade off. However, we note that foundation models are inherently less suitable for low latency inference tasks due to their size. Therefore, for practitioners planning to use foundation models, such as in medical imaging, latency should not be a primary concern from the start. It's important to recognize that in such applications, the challenge isn't inference throughput but rather the capacity to learn a model that can handle high-resolution images effectively.
>
> Moreover, if we solely focus on real-time applications, foundation models typically require distillation or similar techniques to enhance inference latency for feasible deployment. Should distillation be employed, our approach would match the latency of a standalone foundation model.
>
> 2 - We used the features from the last layer, i.e., before the linear classifier. We will clarify this in the manuscript. Early on in development, we experimented with trimming parts of the foundation models to generate the offline features. Using the DDSM dataset, we found that keeping only the first 5 blocks of the Dinov2-base model to generate the embeddings yielded a slight performance boost. We did not further investigate in this direction since choosing the layer to utilize would be another hyperparameter to select with arguably limited returns. Regardless, we agree with the reviewer that the impact of hierarchically less abstract features on LOFFTA is interesting and we will add an ablation on trimming foundation layers to our work.
>
> 3 - The performance boost due to pooling may seem surprising but we believe pooling helps the model better focus on features pertinent to the classification task by suppressing the noise inherent in the high dimensional feature space. The benefits of pooling is also another indicator for the robustness of the foundation model features.
>
> 4 - The projection layer is trained end-to-end. We will make the necessary corrections in Eqs. 2 & 4 such that the argmin operation is over both $\theta$ and the parameters of $\Phi_{proj}$.

---

> > ### Comment · Reviewer_AUq9 · 2024-08-12
> >
> > I thank the reviewers for their comments and addressing some of my comments on the weakness. I appreciate the clarifying comments. I would like to follow up on one of their comments:
> >
> > > Therefore, for practitioners planning to use foundation models, such as in medical imaging, latency should not be a primary concern from the start. It's important to recognize that in such applications, the challenge isn't inference throughput but rather the capacity to learn a model that can handle high-resolution images effectively.
> >
> > > Moreover, if we solely focus on real-time applications, foundation models typically require distillation or similar techniques to enhance inference latency for feasible deployment. Should distillation be employed, our approach would match the latency of a standalone foundation model.
> >
> > Agree that accuracy might be more important than inference latency in medical applications. But I don't agree with the fact that LOFF-TA is necessarily a better learner than a simple linear. The way I see it if you put a few a few more trainable ResNet blocks beyond the last frozen layer, would it probably match the accuracy LOFF-TA? i.e. extracting out the improvements by the tensor augmentation + projection layer, versus a ResNet block + projection block. I feel like they are similar learners. Yes the augmentations are a nice addition in a unique feature space. But it would justify changing my rating.  I would like to keep my rating but I definitely think this is interesting work the authors should build upon.

---

> > > ### Author Response · Authors · 2024-08-13
> > >
> > > There may be a misunderstanding regarding the results.
> > >
> > > > But I don't agree with the fact that LOFF-TA is necessarily a better learner than a simple linear.
> > >
> > > The opposite is what we have shown empirically in our main results, in Table 1. LOFF-TA is consistently better than a simple linear classifier, i.e. the first rows of our tables.
> > >
> > > > The way I see it if you put a few a few more trainable ResNet blocks beyond the last frozen layer, would it probably match the accuracy LOFF-TA?
> > >
> > > We already give an answer to this criticism in the “Frozen+DeiT-S” rows in our Table 2 (not with ResNet layers, but with DeiT layers). LOFF-TA results are generally on par with such a setup, but LOFF-TA is far less computationally demanding. More importantly, this setup still prevents training with high resolution images due to prohibitive computational costs.
> > >
> > > > Yes the augmentations are a nice addition in a unique feature space. But it would justify changing my rating. I would like to keep my rating but I definitely think this is interesting work the authors should build upon.
> > >
> > > We thank the reviewer. However, based on the stated weaknesses and our discussion, we do not think their comments justify their rating. We kindly ask them to reconsider if our paper warrants a “3” which according to the guidelines indicates “technical flaws”, “weak evaluation”, “inadequate reproducibility” or “incompletely addressed ethical considerations”.

---

### Official Review · Reviewer_1T2n · 2024-07-12

**Soundness:** 4
**Presentation:** 4
**Contribution:** 3
**Rating:** 6
**Confidence:** 4

**Summary:**

This paper introduced a training scheme considering offline foundation features with tensor augmentations LOFF-TA, focusing on limited resource settings. They basically trained a classifier on cached features from frozen foundation models with an augmentation process to these cached embedded features. Their proposed training mechanism is claimed to speed up the training process by $37\times$ with appx. $26\times$ reduction of GPU memory usage. The authors validated their approach on eleven image classification benchmarks and showed similar performances as standard image augmentations and fine-tuned foundation models.

**Strengths:**

**S1. Simple yet rich architecture.** I appreciate the authors' approach to augment the frozen foundation model features that are being used to train a compact classifier resulting in solid training time and memory space improvement.

**S2. Strong experiments and ablation studies.** The authors validated their model on eleven image classification benchmarks under ViT-B and ViT-G backbones. The ablations studies showed the effectiveness of tensor augmentations and their variances, pooling operation under different network sizes, and presence of [foundation CLS, layer norm]. The results show promising improvements over the unfrozen-linear method under all the network backbones.

**S3. Paper writing and presentations.** The paper is very well written. I like their overall presentation, justifications, and motivations.

**Weaknesses:**

**W1. Adaptation of spatial augmentations.** For high-resolution images, performing augmentations on cached embedded features highly depends on the size of the latent space from where the offline foundation features are being cached. Typically in the medical image domain where one needs to deal with very high-resolution images/volumes (e.g., $256\times256\times256$), it is necessary to consider the topological features when we generate/augment images from raw data that should not change the image class. In that case, where the authors are performing spatial augmentations on the latent features, there might be scenarios where the topological features get distorted from the flip or the classes change from other spatial augmentation operations. Though I understand that spatial augmentations might be beneficial in terms of natural images, these kinds of augmentations might suffer from experimenting on high-dimensional medical images.


**W2. LOFF-TA generalizability on high dimensional 3D volumes.** I appreciate the authors for experimenting with diverse 2D datasets ranging from natural to medical images. However, in the medical domain, we typically follow 3D volumes for complex disease diagnoses such as classifying tumors, neurodegenerative disease, myocardial infarction, etc., which are considered to be high dimensional, e.g., $128^3$ or $256^3$, where it is pretty complex to preserve anatomical information by augmenting on low-dimensional latent features. Experimenting with this type of application on datasets like OASIS, ADNI, ACDC, Synapse multi-organ, etc. would further strengthen the argument about offline tensor augmentations on high-dimensional medical images.


**W3. Pictorial latent space depictions.** To validate the proposed model's efficiency compared to image augmentations, the authors might want to visualize the latent feature spaces on both image/tensor augmentations. This would further ensure that offline latent tensors preserve the important spatial features similar to the latent image features. After that, augmenting the offline tensors would make more sense in the carried-out image analysis tasks.

**Questions:**

Please see the weaknesses section. I tried to discuss all the findings and questions there. Following, please find some of my concerns and suggestions the author might want to rebut in the rebuttal

1. The validation of their model on high-dimensional medical images is missing. The author might want to show very brief experiments on how their proposed approach performs on 3D medical classification tasks.

2. How the offline tensor augmentations preserve the anatomical features compared to the online tensor/image augmentations? Consider the small anatomical features such as ventricles that are very important when we experiment on finding the presence of neurodegenerative disease from 3D brain MRIs.

**Limitations:**

The authors carefully addressed the limitations and societal impact of their work.

---

> ### Author Rebuttal · Authors · 2024-08-06
>
> We thank the reviewer for their comments.
>
> **Weaknesses**
>
> W1 - We do not think spatial tensor augmentations will harm anatomical features more than voxel space augmentations would. However, some care must be taken in the selection of the tensor augmentations. In Appendix C, we pointed out that common sense insights from the 2D pixel space carries over to the tensor space, e.g. applying vertical flips to a building facade classification task is harmful, both in image and tensor spaces. Surprisingly, we noted that the harmful impact of unsuitable augmentations are somewhat mitigated in the tensor space and we believe a similar phenomenon might occur when it comes to voxel space augmentations.
>
> W2 - We agree with the reviewer that demonstrating how our method performs on tasks with 3D volumes would be valuable. We will add these experiments to our study.
>
> W3 - We will include this to our study. We note that we made a related point in our discussion. Due to the non-linearity of the foundation models, applying a vertical flip in pixel space and then generating the embedding will be different from generating an embedding and then applying a tensor space vertical flip. However, we believe the resulting tensors in both scenarios will be visually alike (after dimensionality reduction), as those we provided in Appendix C. Thus, we believe the structures will be preserved, but some small numeric variations will be observed.
>
> **Questions**
>
> 1 - We will add these experiments to our study.
>
> 2 - We believe our approach will not harm the anatomical features more than feature extraction by using a foundation model does. We will empirically validate this.

---

### Official Review · Reviewer_TzTJ · 2024-07-12

**Soundness:** 3
**Presentation:** 3
**Contribution:** 2
**Rating:** 5
**Confidence:** 3

**Summary:**

The authors propose to store low dimensional representations of images that are obtained after passing them through pretrained (foundation) models. Only afterwards augmentation strategies ("tensor transformations") for training the downstream classifier are employed. This has the benefit that the foundation model is not part of the training processm, which is computationally cheaper. More importantly, some dataset specific optimization is obtained, without the need of finetuning the complex / expensive foundation model itself.  The authors present a set of relevant tensor transformation and compare against the baseline.

**Strengths:**

* The authors follow an interesting idea - using (and augmenting) offline features rather than full images for training.
* Using the foundation model as a "feature extractor" is interesting. Even more interesting is identifying a set of operations that mimick standard data augmentation in the resulting feature / tensor domain
* Results are promising. To the best of my knowledge the concept of the tensor transformation is novel.
* The paper is well written

**Weaknesses:**

* It would have been interesting to study tensor augmentations in a more systematic / more formal way
* I am not sure why this approach would be particular suited for high-resolution images. The authors describe how foundation model optimized for large images can be used to deal with small images via the offline eature / tensor augmentation approach. This implies that the presented method is particulary well suited deal with_low-resolution_ images, but not the opposite, isn't it?
* Using a pretrained (foundation) model as a feature extractor is not necessarily an overly innovative idea.

**Questions:**

* See above

**Limitations:**

All very well done.

---

> ### Author Rebuttal · Authors · 2024-08-06
>
> We thank the reviewer for their comments.
>
> 1- We have made efforts to choose our wording carefully and clearly outline the limitations of our study. We welcome any suggestions for improvement and would be grateful if the reviewer could identify specific sections of the manuscript that might need further refinement.
>
> 2 - LOFFTA caches foundation model embeddings using a single forward pass through the dataset. We apply pooling before storing the embeddings, then load them and apply tensor augmentations on reduced dimensional representations to train a small classifier. When the input image resolution increases, the GPU memory requirements normally increase quadratically, mostly due to gradient computation. Since we store embeddings from a single forward pass without gradient computation, it's feasible to use higher resolution images for this purpose. However, as noted in Table 1, using these same high-resolution images for regular fine-tuning is not practical in settings with limited computational resources.
>
> We know that increasing the resolution improves performance in some tasks, e.g. detecting small tumors becomes easier in higher resolutions. The computational efficiency and reduced GPU memory requirements of our method enables practitioners to freely use images with higher resolutions, making it particularly suited for high resolution images. We will clarify the manuscript regarding this topic.
>
> 3 - To the best of our knowledge, we are the first to employ caching in this context. We believe the simplicity of our idea is a benefit (in addition to its efficiency). However, we are happy to cite previous work that we have missed if the reviewer can point those out.

---

### Official Review · Reviewer_PGYR · 2024-07-14

**Soundness:** 2
**Presentation:** 3
**Contribution:** 3
**Rating:** 4
**Confidence:** 5

**Summary:**

This paper proposed a framework to efficiently use foundation features for online serving. The idea (offline feature extraction from foundation model and online inference with lightweight model) is simple but interesting/effective.

**Strengths:**

1. The paper is well written and easy to follow.
2. The paper proposed a simple yet effective framework on how to efficiently use visual foundation model features for online serving.
3. The experimental results demonstrate the proposed method is promising.

**Weaknesses:**

1. Can you please list the classification accuracy of the raw visual foundation models (VFM)? If the performance of the raw VFM is high, we can directly output the probability score and save them in a database instead of extracting features offline and then do online inference with a followup lightweight model.
2. I'm concerned the potential application scenarios of the proposed scheme is limited, since it may only be able to be leveraged on image classification scenarios.
3. When the authors claim efficiency improvement, the baseline is somewhat unfair, because the proposed methods actually need an offline feature extraction step which would also consume computation resources.
4. The tensor augmentation is actually limited to several spatial augmentations, while there are many for raw image classification, for example, color-jittering and mixup..

**Questions:**

1. I agree the tensor augmentation is reasonable for image embeddings (such as flip, rotate, translate and resizing). Is there any way to expand the the tensor augmentation w.r.t spatial operators to the video embeddings?

---

> ### Author Rebuttal · Authors · 2024-08-06
>
> We thank the reviewer for their comments.
>
> **Weaknesses**
>
> 1 - Reporting such an accuracy is not possible. Generally, the classes included during pre-training and fine tuning are different, e.g. self-supervised pre-training of Dino vs. supervised fine-tuning on retinopathy images. Furthermore, the output dimensionality of pre-training and fine-tuning tasks would rarely match, thus a direct mapping for accuracy measurement is difficult, e.g., APTOS2019 having 5 classes vs. the very high dimensionality of Dino outputs during pre-training.
>
> 2- “The method is for image classification” cannot be a limitation. We ask the reviewer to consider many valuable studies in the literature only addressing image classification tasks.
>
> 3 - Offline feature extraction is run only once, without computing or storing gradients, and can be done with any batch size. This cost does not affect computational requirements in any meaningful way. Due to the reviewer’s request, we ran a small experiment and found embedding generation to take approximately 1/10th of one fine-tuning run, most of this time being spent on read/write operations on a HDD. Recognizing that in most experiments the bulk of computation is spent on hyperparameter search, the time spent on a *single* no-gradient forward pass through the dataset is not impactful.
>
> 4 - As we discussed in the paper, our results are surprising exactly because tensor augmentations are more limited and less well-researched than image augmentations yet their impact is comparable. We believe our work can pave the way for more research regarding tensor augmentations.
>
> **Questions**
>
> 1 - All of our proposed tensor augmentations can be directly applied to video embeddings along with uniform frame sampling or tubelet embedding.

---

> > ### Comment · Reviewer_PGYR · 2024-08-13
> >
> > Thanks for the authors' response. I agree that it is also very valuable to address image classification tasks. However, if it is only for image classification, the paper should better thoroughly consider the different types of features, i.e., spatial detailed features and rich semantic features. Also, the paper should formerly discuss the augmentation types which can be used in TA and which cannot be used now.

---

> > > ### Author Response · Authors · 2024-08-13
> > >
> > > We are struggling to understand what the reviewer’s criticisms are.
> > >
> > > > the paper should better thoroughly consider the different types of features, i.e., spatial detailed features and rich semantic features.
> > >
> > > We do consider both features, i.e. spatial features are the features we augment (Section 3.3), and rich semantic features (we assume the reviewer is referring to the CLS features), are also included during training with a subsection (lines 257-265) detailing how different inclusion strategies might impact performance.
> > >
> > > > the paper should formerly discuss the augmentation types which can be used in TA and which cannot be used now.
> > >
> > > There must be a typo in the reviewer’s comment. We believe the reviewer means “the augmentation types that could be used in pixel space that cannot be used in tensor space”. We discussed these issues in Lines 158-162, Section 5.3, and Appendix C.

---

### Decision · Program_Chairs · 2024-09-25

**Decision:**

Accept (poster)

**Comment:**

The reviewers agree that the manuscript "Learning from Offline Foundation Features with Tensor Augmentations" (LOFF-TA) presents a valuable and innovative contribution to the field. The proposed approach, which enables efficient training of classifiers using cached embeddings from a foundation model, shows significant potential, particularly in resource-constrained settings. LOFF-TA not only substantially reduces computational and memory requirements but also demonstrates competitive performance across a variety of benchmarks.

The reviewers commend the authors for clarifying the suitability of LOFF-TA for high-resolution images, highlighting that the method’s computational efficiency and reduced memory usage make it especially advantageous in these scenarios. Furthermore, the reviewers appreciate the authors' commitment to extending their work to include experiments on 3D volumes and to visualizing the latent feature spaces. These additions will undoubtedly strengthen the paper’s contributions.

The reviewers encourage the authors to incorporate the proposed revisions and additional experiments into the final manuscript. Doing so will enhance the clarity of the work and further validate the claims made.

In summary, the paper presents a promising approach with the potential to significantly impact the field, especially in environments with limited computational resources. The reviewers recommend accepting this paper, with the expectation that the authors will implement the proposed improvements.